# Decrypting Cryptic Crosswords:
# Semantically Complex Wordplay Puzzles as a Target for NLP

**Joshua Rozner**
Stanford University
rozner@stanford.edu

**Christopher Potts**
Stanford University
cgpotts@stanford.edu

**Kyle Mahowald**
University of Texas at Austin
mahowald@utexas.edu

## Abstract

Cryptic crosswords, the dominant crossword variety in the UK, are a promising target for advancing NLP systems that seek to process semantically complex, highly compositional language. Cryptic clues read like fluent natural language but are adversarially composed of two parts: a definition and a wordplay cipher requiring character-level manipulations. Expert humans use creative intelligence to solve cryptics, flexibly combining linguistic, world, and domain knowledge. In this paper, we make two main contributions. First, we present a dataset of cryptic clues as a challenging new benchmark for NLP systems that seek to process compositional language in more creative, human-like ways. After showing that three non-neural approaches and T5, a state-of-the-art neural language model, do not achieve good performance, we make our second main contribution: a novel curriculum approach, in which the model is first fine-tuned on related tasks such as unscrambling words. We also introduce a challenging data split, examine the meta-linguistic capabilities of subword-tokenized models, and investigate model systematicity by perturbing the wordplay part of clues, showing that T5 exhibits behavior partially consistent with human solving strategies. Although our curricular approach considerably improves on the T5 baseline, our best-performing model still fails to generalize to the extent that humans can. Thus, cryptic crosswords remain an unsolved challenge for NLP systems and a potential source of future innovation.

## 1   Introduction

Modern computational models have made great progress at handling a variety of natural language tasks that require interpreting rich syntactic and semantic structures [6; 30; 25; 32]. However, in NLP [2; 26; 3] as in other areas of AI [22], machines still lag humans on tasks that require flexible problem solving, rapid learning of unseen tasks, and generalization to new domains. Just as complex games mastered by human experts, such as chess, Go, and video games, have proved a fertile ground for developing more flexible AI [36; 35; 27], we propose that creative language games are a rich area for developing more flexible NLP models. In particular, we argue that linguistic tasks involving meta-linguistic reasoning pose an important and significant challenge for state-of-the-art computational language systems.

One such domain is cryptic crossword puzzles. Cryptics are the most popular style of crossword in the United Kingdom and appear in major newspapers like *The Times* and *The Guardian*. Cryptic clues have two parts: a **definition** and a **wordplay** cipher that, when placed adjacent to each other, read like fluent natural language. For example, consider this NLP-centric cryptic crossword clue: "But everything's really trivial, initially, for a transformer model (4)" with answer "BERT". "(4)" is the **enumeration** and specifies the number of letters in the answer. Solvers must identify which

Table 1: Examples of five common clue types in cryptic crosswords, all demonstrating clues for the answer: BERT. Indicators, where they occur, are italicized. The wordplay substrate appears in bold. Typographical emphasis added to aid reader, only; actual clues have no such indication.

| Clue type | Clue example | Explanation for this example |
|---|---|---|
| **Anagram**: An anagram clue requires scrambling some set of clue letters to produce the answer. | *Confused*, **Bret** makes a language model (4) | *Confused* indicates that we should "confuse" (anagram) the letters of "Bret" to get BERT. |
| **Initialism**: An initialism requires taking the first letters of a phrase | **B**ut **e**verything's **r**eally **t**rivial, *initially*, for a language model (4) | *initially* indicates taking the first letters of "But everything's really trivial". |
| **Hidden**: The answer occurs within a larger phrase. | Language model *in* som**ber t**ext (4) | *in* indicates that a word is hidden inside a phrase. Extract the word BERT from the phrase "somber text." |
| **Charade**: For a charade clue, each part of the answer is clued sequentially. | A language model ex-ist? Right time! (4) | "exist" becomes "BE" since "exist" and "be" are synonyms. A standard abbreviation for "right" is "R." A standard crossword abbreviation for "time" is "T." This clue type does not have an indicator. |
| **Double definition**: In a double definition clue, two synonyms or phrases appear next to each other, each of which can refer to the answer. | Model Sesame Street character (4) | Bert is a valid answer for "Sesame Street character", and it is also a model. Double definitions do not have indicators. |

part of the clue is definition and which is wordplay. The **definition** should be a semantically valid description of the answer word: "a transformer model" can mean "BERT". The **wordplay** part is the remainder of the clue: "But everything's really trivial, initially". The word *initially* in this context is used as an **indicator**, which tells the solver to take the *initial letter* of each of the preceding words (**b**ut **e**verything's **r**eally **t**rivial) to produce "BERT". Because both the wordplay and the definition give the same 4-letter word (which is what the **enumeration**, "(4)", calls for), we can be confident that "BERT" is the correct answer.

The clue just discussed is an example of the *initialism* clue type, which is one of roughly 15 major clue types. Other types can require anagramming words, finding words hidden in longer phrases, performing synonym substitutions, substituting a word for a soundalike (e.g., "hiring" for "high ring"), or composing a number of these manipulations. See Table 1 for examples of several other clue types with answer "BERT" and Appendix A for examples of actual clues from the dataset. As with American-style clues, cryptics require world knowledge and linguistic flexibility to match the definition, but also considerable attention to meta-linguistic concepts to solve the wordplay. In this paper we study the language task of solving individual cryptic clues, rather than full puzzles since, unlike American-style crosswords, cryptics generally have unique answers and so are less reliant on grid constraints in order to achieve a complete solution.

While cryptics pose a challenge to novice solvers unfamiliar with their structure, experts flexibly combine world, domain-specific, and linguistic knowledge to solve novel clues. Experts know the rules that govern cryptic clues, but they also reason about them flexibly and apply them to solve novel clues. In the psychology literature, it has been claimed that cryptics depend on domain-general intelligence and are an ideal domain for studying the "aha moment" in humans [11; 10] because experts can solve them with high accuracy (but rarely at first glance), they can be easily generated, and they require drawing on a diverse range of cognitive abilities. Therefore, we believe that cryptic crosswords are an excellent domain for developing computational language systems that "learn and think like humans" [22], posing an interesting and important challenge for modern machine learning.

Our main contributions are, first, a cleaned dataset of cryptic crosswords clues from theguardian.com, consisting of 142,380 clues from 5,518 puzzles over 21 years,[1] and second, a novel curriculum learning approach, in which the model is first fine-tuned on related, synthetic tasks (e.g., an augmented word descrambling task) before tackling actual cryptic clues. This method meaningfully improves on a standard T5 seq-to-seq approach and on the best model of Efrat et al. [7]—concurrent work that presents a similar dataset and similar neural baseline using T5.

In this paper, we aim not only to present a dataset and propose a novel solution but also to characterize the problem and motivate its importance. To that end, we elucidate the task with three non-neural baselines and fine-tune T5 [31], a Transformer-based [38] encoder-decoder, as a neural baseline. Since the character-level wordplay inherent to cryptics might be challenging to language models with subword tokenization (T5 uses SentencePiece [21]), we study whether T5 has or can acquire meta-linguistic knowledge. In Section 6.1 we examine whether T5 learns meta-features of the task related to answer length. In Section 6.3 we use a descrambling task to assess whether T5 understands the character composition of words and whether the model can make use of linguistic and meta-linguistic information simultaneously. Our results show, perhaps surprisingly, that the subword-tokenized T5 model is quite robust to character-level challenges. Moreover, the descrambling task may serve as a useful benchmark task in guiding the development of new approaches on the cryptics task.

Given the compositional nature of cryptic clues, we investigate the extent to which the model generalizes under increasingly difficult data splits. In Section 6.2 we introduce a new form of disjoint data split to address T5's robustness to inflection: a word-initial disjoint split that segments clue–answer pairs based on the first two letters of the answer. In Section 6.4 we examine the systematicity of the model's answer generation by perturbing the wordplay portion of anagram clues, showing that its behavior is partially consistent with human solving strategies.

Although our novel curricular approach considerably improves performance on this task, fully solving cryptics remains a challenge for modern machine learning, with expert humans still outperforming machines. Therefore, we hope that our dataset will serve as a challenging benchmark for future work.

## 2   Related work

While there is an existing literature on puns and wordplay in NLP [17; 20; 24] as well as on solving American-style crosswords [13; 23; 34], there has been relatively little work using NLP methods on cryptics, which require processing highly compositional language and disambiguating surface-level from hidden meaning. Hart and Davis [16] laid out a computational framework for solving the problem, and Hardcastle [14, 15] proposed some rule-based solutions. The app Crossword Genius from Unlikely AI solves cryptic clues and gives human-understandable explanations. Because its method is proprietary and not available for open testing, we do not report it as a baseline but note that it is competitive. Deits [5] offers a rule-based solution, which can also output explanations, and which we test on our dataset. Most recently, in work concurrent to ours, Efrat et al. [7] release a similar dataset of cryptic crossword clues and present a neural baseline using T5-large. In Section 6.5, we discuss how our new data split and curricular approach improve on their work.

Our curricular approach fits into the space of recent work on pre-finetuning [37; 1] and curricular approaches for compositional tasks [12; 40]. Our approach is loosely related to the pre-finetuning of Aghajanyan et al. [1] but differs in that our curriculum is composed of fewer tasks that are all closely related to the primary task and synthetically generated. Our approach resembles Geva et al. [12] in that we attempt to endow language models with a specific kind of reasoning by training in a multi-task setup over synthetic data. In the vein of Wu et al. [40], which trains on synthetic datasets to encode inductive bias, our approach can be understood as encoding wordplay functional biases.

Whether large pretrained Transformer models generally pick up on meta-linguistic features like word length and the character composition of words is an open question. Brown et al. [4] explore a set of restricted word unscrambling tasks in the few-shot (versus fine-tuned) setting for GPT-3. Probing results from Itzhak and Levy [18] suggest that information about the character composition of words is present in the embeddings of pretrained, subword-tokenized models. This seems to

---

[1]We release the dataset along with all code to reproduce the results in this paper at https://github.com/jsrozner/decrypt.

confirm our result that subword-tokenized models do have more knowledge of character composition and meta-properties of words than we might have expected.

# 3 Dataset and task

We present a cleaned dataset of 142,380 cryptic crossword clues from 5,518 puzzles published in *The Guardian* from July 1999 to October 2020. We also introduce a challenging "word-initial" disjoint split after observing that T5 is robust to inflection. Overall, the dataset has 55,783 unique answers, giving a mean frequency of 2.55 clues per unique answer. Answers in the dataset consist of one to six words, with most (97.4%) having one (83.3%) or two (14.1%) words. Full details of dataset preprocessing are given in Appendix B, and in addition to releasing the full dataset, we include code to fully replicate our data download, pre-processing pipeline, and split generation in the repository.

## 3.1 Task

We frame the problem as a standard seq-to-seq task from inputs (clues with length enumeration) to outputs (answers). For example, one input could be *But everything's really trivial, initially, for a transformer model (4)*, with output *BERT*. This is consistent with how clues are presented to human solvers. Full details of the input–output specification for each model are provided in Appendix C, along with other experimental details.

## 3.2 Splits

As motivated in the introduction (and in Section 6.2), we consider three splits. The **naive (random) split** is a standard 60/20/20 random split into train, dev, and test. The **disjoint split** ensures that all clues with the same answer appear in only one of train, dev, or test. For example, if any clue for which the answer is "BERT" is in the train set, then *all* clues for which the answer is "BERT" are in the train set. The disjoint split is used to test composition and generalization. It prevents an approach that relies only on lexical similarity across clues (like KNN) from succeeding on the task. Finally, the **word-initial disjoint split** is designed to address T5's robustness to inflection. For this split, we enforce that all clues whose answers start with the same two letters will appear in the same set. For example, all clues that have answers starting with 'ab' like "abacus," "abdicate," "abdicates," will be grouped, ensuring that inflections or spelling variations of the same base word occur in a single split.

## 3.3 Metrics

Our primary metric is whether the top-ranked output is correct, *after* filtering to outputs of the correct length. We filter because each clue in a puzzle has a hard length constraint, i.e., a human solver *cannot* pencil in a solution that is of the wrong length. Additionally, we report how often the correct answer is contained in the top 10 outputs after length filtering. This is a useful metric since, when clues are presented in the context of a full puzzle, solvers use information from interlocking answers to narrow down a set of candidates. For instance, the best-performing crossword solver for American-style (non-cryptic) crosswords relies heavily on satisfying grid constraints [13]. Comparing post-length-filter results from Section 4.5 with pre-filter results from Section 6.1, the length filter is seen to increase the top-10 metric by roughly 6% for T5 (with length enumeration given).

# 4 Baseline models and results

To characterize how simpler approaches perform on this task, we test three non-neural baselines: a WordNet-based heuristic model, a k-nearest-neighbor bag of words model (KNN BoW), and a rule-based model designed for the task [5]. For a neural baseline, we fine-tune T5-base [31]. For models that can interpret the answer-length enumeration as a textual token (KNN and T5), we append it to the input string (e.g., "(4)"). For these two models, we report results both with and without appending the enumeration. Implementation details are discussed in Appendix C.

### 4.1 WordNet

Our first baseline is a simple heuristic based on clue structure. It takes advantage of the fact that the definition part of a cryptic clue always appears at the beginning or end of the clue. For instance, in the double definition clue for "BERT" in Table 1, "Model Sesame Street character," the word "model" appears at the beginning and is a definition (in this case, a hypernym) for the answer "BERT". We use WordNet [8], a large database of English word meanings, to build a reverse dictionary via the synonym, hyponym, and hypernym graphs. We take as candidate solutions the set of reverse dictionary outputs for the first and last words of a clue. For example, if "dog" appears at the start of a clue, candidates would include "animal", "canine", "labrador", etc. Ranking outputs by character-level overlap with the rest of the clue slightly improves performance, since answers sometimes are rearrangements of the characters in the wordplay portion of the clue.

### 4.2 KNN BoW

To assess whether clues that are close in lexical space have similar answers, we use a KNN model on top of a bag-of-words featurizer.

### 4.3 Rule-based

Finally, to evaluate how well a rule-based approach, with a hand-coded grammar, can perform on the task, we use the Deits [5] solver.[2] This solver handles anagrams, initialisms, substrings, insertions, and reversals in a rule-based way. While the rule-based version includes common clue types, an inherent limitation of this approach is that it is difficult to enumerate all possibilities. For instance, the Deits solver does not include charade-type clues, nor double definitions. Moreover, the rule-based solver uses WordNet's [8] word similarity functionality to rank outputs, meaning that, in general, it will fail on clues that have definitional parts consisting of more than one word (e.g. "language model" from our example in the introduction would not be matched).

### 4.4 T5: vanilla seq2seq

For our baseline neural seq-to-seq approach, we fine-tune the Transformer-based [38] T5-base model [31], starting from HuggingFace's [39] pretrained model parameters. T5 is an encoder-decoder language model pretrained on the C4 corpus [31]. Fine-tuning is done via supervised learning (teacher-forcing) over standard seq-to-seq input-output pairs. At test time, we generate outputs using beam search. As described in Section 3.3, we filter the outputs to those of the correct length and evaluate by checking whether the top result is correct and whether the answer occurs in the top 10. See Appendix C for details, including hyperparameter selection.

### 4.5 Results

In Table 2, we report metrics for dev and test sets on both the naive (random) split and word-initial disjoint split (discussion on disjointness in Section 6.2). While the WordNet baseline achieves some success (2.6% and 10.7% top-1 and top-10 on the test set), it is inherently limited, since it cannot handle clues with multiword definitions and lacks a good ranking mechanism. KNN does better, achieving 6.1% and 11.3% with length enumeration. The rule-based solver achieves 7.3% and 14.7%, marginally outperforming the KNN baseline. Though our T5 neural baseline outperforms all non-neural baselines, achieving 16.3% and 33.9%, it leaves considerable room for improvement.

## 5 Curriculum learning

Solving cryptic clues uses different linguistic and reasoning abilities compared to the natural language tasks on which T5 is pretrained. Thus, during fine-tuning on the cryptics task, the model must learn many sub-parts of the problem simultaneously: how to look up definitions, how to identify wordplay indicators, how to perform wordplay operations, etc. Although these elements of the task

---

[2]Deits [5] has a more recently implemented solver in Julia that was used by Efrat et al. [7]. We use the Python version, which may have small differences from the Julia version and is reportedly much slower (see Appendix C for more details).

Table 2: Results for baselines and top curricular approach. Details on the curricular approach are given in Section 5. Metrics are percentages calculated over the top ten model outputs, after filtering to outputs of correct length.

| Model | Naive (random) split | | | | Word-initial disjoint split | | | |
| | Top correct | | Top 10 contains | | Top correct | | Top 10 contains | |
| | dev | test | dev | test | dev | test | dev | test |
|---|---|---|---|---|---|---|---|---|
| WordNet | 2.8 | 2.6 | 10.8 | 10.7 | 2.6 | 2.6 | 10.6 | 10.5 |
| Rule-based | 7.2 | 7.3 | 14.8 | 14.7 | 7.4 | 7.3 | 14.9 | 14.5 |
| KNN (no lengths) | 5.6 | 5.6 | 9.9 | 10.1 | 0.0 | 0.0 | 0.0 | 0.0 |
| KNN (lengths) | 6.0 | 6.1 | 11.2 | 11.3 | 0.0 | 0.0 | 0.0 | 0.0 |
| T5 (no lengths) | 15.3 | 15.6 | 29.4 | 30.0 | 0.9 | 1.0 | 4.8 | 5.1 |
| T5 (lengths) | 16.0 | 16.3 | 33.1 | 33.9 | 1.1 | 1.1 | 5.6 | 5.8 |
| Curricular: ACW + ACW-descramble | **21.5** | **21.8** | **42.2** | **42.4** | **6.1** | **6.5** | **18.9** | **20.0** |

each individually benefit from T5's natural language pretraining, our standard T5 baseline suggests that learning to compose them all at once is challenging. We show that a curriculum of synthetic datasets can address this, substantially improving performance on the primary cryptics task. We test a number of different curricular tasks and task sets and discuss which are most effective. This curricular approach may be useful in other settings where focused linguistic capabilities are required.

## 5.1 Curricular datasets

The logic of our curricular approach is to provide some guidance on the sub-parts of the problem space before building up to fully compositional cryptic clues. For curricular training, we create four datasets designed to improve performance and elucidate what makes a good curricular task for this problem. We process a public American crossword clue dataset [29], henceforth "ACW-data" for "American Crossword", and generate three datasets: ACW, ACW-descramble, ACW-descramble-word. After preprocessing, ACW-data has 2.5m clue–answer pairs with 250k unique answers, giving a mean frequency of roughly ten clues per unique answer. Unlike cryptic clues, American crossword clues often involve relatively straightforward synonym or definition substitions, so the ACW dataset can be used to train definition lookup.

We also produce a separate anagram dataset from a publicly available English dictionary. Details to produce all datasets are included in Appendix D.1. In all example clues that follow, the target output is "PETAL" and we use labels (prepending a word and colon) to help the model distinguish tasks.

1. **ACW**: ACW-data dataset in input–output form (i.e., a seq-to-seq version of American-style crossword clues) with no augmentation. For example, *phrase: flower part (5)*.

2. **ACW-descramble**: For each clue–answer pair in ACW-data, we create an input that models a cryptic clue by scrambling the answer word and prepending or appending it to the clue portion. For example, we scramble "petal" and randomly place it at the beginning (*descramble: etalp flower part (5)*) or end (*descramble: flower part etalp (5)*) of a clue.

3. **ACW-descramble-word**: A seq-to-seq task that is just the descrambling of answer words. When compared to ACW-descramble, this elucidates the importance of curricular and primary task similarity, in particular whether teaching a bare descrambling task is helpful to the model. Example: *descramble word: etalp (5)*.

4. **Anagrams**: Using a dictionary as starting point, we synthesize an anagram dataset: we pair a word (to be anagrammed) with an anagram indicator (e.g., "mixed up", "drunk", "confusingly") and ask the model to produce a valid anagram (i.e., a scrambled version of the word that is itself also a valid word). For example, *anagram: confusingly plate (5)* (rearrange the letters of 'plate' to get "PETAL"). The anagram dataset simulates the anagram type of wordplay in cryptic clues, with definition part omitted.

Table 3: Curricular results (left) and sample metrics for meta-linguistic feature analysis (right)

(a) Curricular approaches on the naive (random) split. Metric is correctness of top-output (5 beams with length filter).

| Curricular dataset | Percent correct full dev set | anagram subset |
|---|---|---|
| Baseline (no curricular) | 15.7 | 13.7 |
| ACW | 18.3 | 14.4 |
| ACW-descramble | 13.1 | 21.4 |
| ACW + ACW-descramble | **20.2** | 24.0 |
| ACW + ACW-descramble-word | 17.8 | 18.3 |
| ACW + anagram | 17.1 | 19.1 |
| ACW + ACW-descramble + anagram | 20.1 | **27.1** |

(b) Sample metrics calculated over top 10 outputs *without* length filter, using naive split.

| Model | % sample contains answer (top-10, no filter) | | % outputs with correct length | | % outputs correct word count | |
|---|---|---|---|---|---|---|
| | dev | test | dev | test | dev | test |
| KNN | | | | | | |
| – (no lengths) | 6.5 | 6.6 | 13.4 | 13.3 | 70.7 | 70.7 |
| – (lengths) | 10.6 | 10.7 | 85.4 | 85.3 | 89.7 | 89.6 |
| T5-base | | | | | | |
| – (no lengths) | 19.0 | 18.8 | 16.0 | 16.2 | 74.2 | 74.1 |
| – (lengths) | 27.5 | 28.1 | 48.3 | 48.5 | 97.9 | 97.9 |

## 5.2 Methods

Our curricular approach is as follows: using the same methods as in Section 4.4, first, we fine-tune T5 on one or more supplementary tasks. Second, we continue fine-tuning on the primary cryptics task, periodically showing batches from the supplementary task(s), to decrease the likelihood of catastrophic forgetting [9]. Full details of our curricular training setup are in Appendix D.2.

## 5.3 Results

In Table 3a, we report results for our curricular approach. Overall, we find that curricular training using ACW + ACW-descramble is best and report in Table 2 a primary task improvement from 16.3% to 21.8% on the random split and from 1.1% to 6.5% on the word-initial disjoint split.

We begin by testing a curriculum with only ACW, which corresponds roughly to teaching the definitional lookup. Observing that ACW improves performance over the baseline, we test ACW-descramble, which is ACW augmented with a scrambled word component (an implicit wordplay). Surprisingly ACW-descramble leads to a decline in performance relative to both ACW and to Baseline. On the other hand, combining ACW + ACW-descramble leads to our best performing approach, demonstrating that a particular curricular combination can improve over two separate tasks. We also compare ACW + ACW-descramble to ACW + ACW-descramble-word. The drop in performance suggests that inputs that are more similar to the final task are more useful than, e.g., a bare descrambling task. To isolate the effect of task choice, all curricular approaches use the same data distribution, train for the same number of curricular fine-tuning steps, and are reviewed during primary training at the same frequency.

Finally, in order to explore whether the curricular training improves performance across the board or just on the clue types directly involved (e.g., anagrams), we report performance on an algorithmically-labeled anagram subset ("anagram subset" column in Table 3a). Adding the Anagrams subtask to the curriculum improves anagram performance but interestingly does not improve performance compared to the top curriculum, ACW + ACW-descramble.[3] We see a similar gain in anagram performance (but drop in overall performance) when training with only ACW-descramble. This suggests that pretraining for a particular clue type can improve performance on that type, but perhaps at the expense of performance on other clue types.

Beyond providing guidance to T5 on the problem sub-parts, this approach also partially addresses the disjointness of train and test sets. For the word-initial disjoint split, by periodically refreshing the curricular task, we remind T5 to produce outputs that are not in the training set of the cryptic split.

---

[3]The distribution and size of the Anagrams dataset is different from the ACW datasets, so we separate curricula involving the Anagrams dataset in Table 3a.

# 6 Model analysis

## 6.1 Learning meta-linguistic properties: output length and number of words

This task requires not just linguistic but also *meta-linguistic* reasoning. We investigate how model behavior changes when the enumeration (the specification of the length of the answer) is appended to the end of the input string as a number in parentheses. Whether large pretrained transformer models generally pick up on meta-linguistic features like word length is an open question.

To study whether the models learn length information, we report how often the top-10 candidate outputs for each clue are the correct length and have the correct number of words *before* applying any length filter. For both the KNN and the T5 models, we find that including the length enumeration improves overall performance, as can be seen in Table 2. (We omit the WordNet and rule-based approaches from this discussion since they have no capacity to learn the meaning of the length enumeration.)

In columns 3 and 4 of Table 3b we see that both KNN and T5 pick up on length information, generating more outputs of the correct length when the enumeration is provided. T5 is particularly proficient at producing the correct number of words in outputs. (Recall that multiword answers are indicated with an enumeration that has two numbers separated by a comma, as in "(4, 6)", indicating an answer like "ALAN TURING".) Given that T5 produces 97.9% of outputs with the correct number of words, it seems plausible that the model is learning a correspondence between the enumeration and the presence of a space or spaces in its output.

## 6.2 Disjointness

Based on the performance of the KNN model on the naive data split, we see that some clues can be solved by picking up on lexical similarity to other clues. Thus, we investigate whether T5 is also picking up on similarity to previously seen clues or if it is learning something about the compositional and functional structure of the cryptic clues.

To assess how much a Transformer-based model like T5 relies on having seen similar clues for the same word, we segment performance on the random split by whether the answer was in the train set. In Table 4a, we see that performance drops from 16% on the full dev set to only 3.0% on the clue subset not seen during training, confirming our intuition that lexical similarity between clues with the same answer plays a role in model success.

To formalize this result, we create and train on the two disjoint datasets described in Section 3: the basic disjoint and the word-initial disjoint splits. The T5 model achieves 3.3% accuracy on the basic disjoint split (dev) and only 1.1% accuracy on the word-initial disjoint split. The drop is likely partially attributable to robustness to inflection, since inflected forms often start with the same two letters.

## 6.3 Wordplay: minimal task

We investigate the extent to which T5, which uses SentencePiece tokenization [21], can perform wordplay-esque tasks like descrambling. We run an experiment as follows: we start with the ACW dataset from Section 5, further restrict to outputs with targets in a dictionary (i.e., no multiword answers), and downsample to 180k clues (10%). We create two descrambling tasks. The first is a direct descramble task, where the input is a scrambled version of the target word (e.g., *etalp* for target *petal*). The second task is a descramble with phrase tasks, in which we append the clue of the clue-answer pair after the scrambled answer letters (e.g., input is *etalp | flower part* for target *petal*). The second task is designed to mimic the cryptic setup, where we have a wordplay (in this case, the implicit descramble function) whose output is conditioned on a (possibly phrasal) synonym. See Appendix E.1 for more details.

We present results in Table 4b. We observe that the model reasonably learns the task on a random split (63.8%) but fails on a word-initial disjoint split (3.7%). Notably, including a phrasal definition alongside the scrambled letters in the input improves outcomes, suggesting that the model is simultaneously incorporating both meta-linguistic character-level and overall word-embedding information. This task can serve as a benchmark for models to solve the cryptic task, since it roughly upper-bounds how well a model can solve wordplays.

Table 4: Disjointness results (left) and descrambling results (right)

(a) T5-base performance (% for top-10 outputs after length filter) on naive, subset of naive not seen in train, disjoint, and word-initial disjoint splits.

| Dataset | Top correct | | Top 10 contains | |
|---|---|---|---|---|
| | dev | test | dev | test |
| Naive (random) split | | | | |
| – Entire split | 16.0 | 16.3 | 33.1 | 33.9 |
| – Subset not in train | 3.0 | 2.8 | 9.5 | 9.7 |
| Disjoint splits: | | | | |
| – Naive disjoint | 3.3 | 3.2 | 12.6 | 12.9 |
| – Word-initial disjoint | 1.1 | 1.1 | 5.6 | 5.8 |

(b) Descrambling task, with and without phrasal definition. Metric is % for top ten outputs without length filter.

| Split and task | Top correct | Top 10 contains |
|---|---|---|
| Random split | | |
| – Descramble | 63.8 | 91.6 |
| – Descramble w/ phrase | 79.4 | 91.2 |
| Word-initial disjoint split | | |
| – Descramble | 3.7 | 12.5 |
| – Descramble w/ phrase | 12.4 | 24.4 |

Given that we observe a considerable drop from the random to disjoint data splits, we test whether T5 can learn the identity function under a disjoint split. We find that the model achieves 100% accuracy on a direct copy task in which the model is trained to simply output the string that is given as input. This suggests that T5's meager performance for descrambling on the disjoint split is not due to an inability to generate an output that has not been seen in fine-tuning.

## 6.4 Assessing systematicity

We investigate the extent to which the model's behavior is consistent with the compositional and systematic nature of human solving strategies. Consider our anagram clue from Table 1: "Confused, Bret makes a language model (4)," for which the answer is "BERT". Using a publicly available list of 18,000 first names, we algorithmically identify clues in the dev set that require the solver to compute an anagram of a first name, and we use those clues to compose two specialized test sets. In the first set, we scramble the original letters (e.g., *Bret* becomes *Treb*). If the model is behaving optimally, performance on this set should not suffer: what is important about *Bret* in this clue is not its semantics but its characters. In the second set we substitute another name of the same length (e.g., *John* for *Bret*). We pick first names because swapping a name does not change the grammaticality or linguistic validity of clues. Here, for a human solver, we would expect a major hit in performance since the correct answer is a valid anagram of *Bret* but not of *John*, and so the clue is no longer a valid cryptic clue. It does, however, still have a valid definition part ("a language model"), and so, if the model is picking up only on the definition part and ignoring the wordplay part, it might still do well. See Appendix E.2 for more task details.

On this set of clues, prior to any modification, we find a baseline accuracy of 33.3%. When scrambling the name, accuracy drops moderately, to 25.2%. When substituting a name with different letters, accuracy falls to 7.0%. This difference in performance on the substitution and scrambling tasks suggests that the model is, to some extent, correctly picking up on the need to use the letters of the name as the wordplay substrate. This is confirmed by the average character overlap between the altered word and the generated candidate answers. We observe 51.2% multiset overlap for the baseline (name unchanged from original clue), 51.0% for substitution of names with the same letters, and 31.4% when substituting names with different letters. For our example clue, this means that we would expect high character overlap between *Bret* and the candidate answers in the baseline set, but high overlap between *John* and the candidate answers in the substitution test set. These results suggest that, like human solvers, the model is sensitive to character manipulations at the location of the wordplay substrate.

## 6.5 Comparison to Efrat et al.

In contemporaneous work, Efrat et al. [7] present a dataset of cryptics from two other major newspapers and fine-tune T5-large for the task. While Efrat et al. [7] conclude that train/test disjointness is important, they do not fully consider T5's robustness to plural and other inflections. The word-initial disjoint split that we present addresses this. In particular, their 'naive' split is the same as our naive split, and their 'official' split is the same as our (naive) disjoint split. To demonstrate that our split is

Table 5: Performance of T5-large as reported by Efrat et al. [7], in our replication of their work, and with our top curricular approach (ACW + ACW-descramble). Metric is correctness of top output (5 beams without length filter) on test set.

| Split | Efrat et al | Our replication of Efrat et al | Top curricular |
|---|---|---|---|
| Efrat 'naive' (test) | 56.2 | 53.2 | 52.1 |
| Efrat 'official' (test) | 7.6 | 10.9 | **19.9** |
| Word-initial disjoint (test) | – | 4.9 | **12.8** |

relevant to the Efrat work, we replicate their results (we train T5-large and report the same metric, correctness of top output with b=5 beams), show a considerable decline in performance under the word-initial disjoint split (10.9% to 4.9%), and finally demonstrate that our curricular approach substantially improves results on the 'naive-disjoint' (10.9% to 19.9%) and word-initial disjoint splits (4.9% to 12.8%). Performance on the 'naive' split does not change considerably with our curricular approach. Results are in Table 5, and further training details are given in Appendix E.3.

## 7   Conclusion

In this paper we introduce a dataset of cryptic crossword clues that can be used as a challenging new benchmark task and develop a novel curricular approach that considerably improves performance on the benchmark. We further characterize the problem with three non-neural baselines and provide methods for investigating model behavior, including a simple descrambling task and an experiment that explores what T5 learns about compositional task structure. Lastly we introduce a challenging word-initial datasplit to evaluate a model's ability to achieve compositional generalization. These contributions demonstrate why this task is worth studying and how it may be relevant to related problems. For example, our curricular approach may be useful in other settings where focused linguistic capabilities are required.

Pretrained contextual embedding models like T5, which draw on a rich base of lexical and linguistic knowledge from pretraining, are a promising candidate for the type of flexibility needed to solve this sort of puzzle. However, T5 does not initially succeed at the task, and although our curricular approach considerably improves task performance, cryptics remain an unsolved problem. Although one might initially think that a character-level tokenization scheme would be necessary for this task, Sections 6.1 and 6.3 suggest that T5 *can* unscramble words (under appropriate generalization splits) and *does* learn a correspondence between a word's tokens and the word's total length.

Given the success of our curricular approach, future work might combine new synthetic datasets under a learned curriculum schedule. In any case, an approach that fully solves this problem will need to more flexibly learn different kinds of wordplay functions and how to functionally compose them to produce the answer word. In that sense, we believe that the cryptic crossword task serves as a good benchmark for those interested in building NLP systems that can apply linguistic and meta-linguistic knowledge in more creative, flexible, and human-like ways.

## Acknowledgments and Disclosure of Funding

This work was supported in part by a Stanford HAI Hoffman–Yee grant.

We thank Saul Pwanson for guidance on the release of crossword-related datasets and William Tunstall-Pedoe for discussions on cryptics and for testing his proprietary solver on our dataset. We thank Armando Solar-Lezama, Josh Tenenbaum, Osbert Bastani, and other members of the Neurosym group for helpful feedback. We especially thank Dana Angluin for suggesting (in 2014!) cryptic crosswords as an undergraduate thesis topic for Joshua Rozner.

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
