# A Clue examples from dataset

Table 6: Examples from our dataset, taken from the train portion of the naive split. Replicates Table 1 in the main paper. Indicators, where they occur, are italicized. The wordplay substrate appears in bold. Typographical emphasis added to aid reader, only; actual clues have no such indication.

| Clue type | Clue example | Explanation for this example |
|---|---|---|
| **Anagram**: An anagram clue requires scrambling some set of clue letters to produce the answer. | Honour, **Ben** and **Noel** *with a new order* (4) | *with a new order* indicates that we should re-order (anagram) the letters of "Ben" and "Noel" to get ENNO-BLE. |
| **Initialism**: An initialism requires taking the first letters of a phrase | *Initially*, **i**s **d**octor **e**lated **at** result of brain operation (4) | *initially* indicates taking the first letters of "is doctor elated at", which gives IDEA, the "result of brain operation". |
| **Hidden**: The answer occurs within a larger phrase. | Cryptic advice f**or a cl**ever *solver to extract* (6) | *solver to extract* indicates that a word is hidden inside a phrase. Extract the word ORACLE from the phrase "for a clever". |
| **Charade**: For a charade clue, each part of the answer is clued sequentially. | Nitrogen and oxygen shown to exist to student chemist (5) | "Nitrogen" becomes "N", "oxygen" becomes "O", "shown to exist" becomes "BE" since they are synonyms, and a standard abbreviation for student is "L" for learner. NOBEL was a chemist! This clue type does not have an indicator. |
| **Double definition**: In a double definition clue, two synonyms or phrases appear next to each other, each of which can refer to the answer. | Painful withdrawal, having raw meat (4,6) | "COLD TURKEY" means both "Painful withdrawal" and "raw meat". Double definitions do not have indicators. |

# B Cryptics dataset preprocessing

To produce the clean dataset, we remove 15,591 clues that interact with other clues in the same puzzle as follows:

1. 7,687 clues that are explicitly marked as being part of a clue grouping (i.e. clues that the puzzle author has explicitly marked as interacting). For example, from Guardian puzzle 21633:[4]

   (a) 20-across: *this cast no blight on semi-conventional party (8,8)*
       "SCOTTISH NATIONAL"

   (b) 5-down: *see 20*

   In this case, the answer must be written into two locations (20-across and-5 down). The first part (20-across) is a valid clue for our models, but we omit clues of this type because programatically parsing them would require simultaneously looking at multiple clues during preprocessing.

2. 607 "continuation" clues or clues that are part of an annotated grouping: These include clues that start with an asterisk (indicating clue grouping) or those that start with an ellipsis, which indicates continuation from a previous clue. For example, from the same puzzle:

   (a) 23-across: *drunken kilty whams a dram ... (4,6)*
       "MALT WHISKEY"

   (b) 24-across: *..and another, by the sound of it, on a 20 isle, while ... (4)*
       "RHUM"

---

[4]Each puzzle can be accessed at `https://www.theguardian.com/crosswords/cryptic/puzzle_id`.

Solving 24-across requires having seen 23-across. ("Rum" is a type of malt whiskey that sounds like Rhum, which is a Scottish isle.) Note that we also needed to substitute Scottish for 20, from 20-across.)

3. 7,066 clues that contain a numeral. Many clues with a numeral are references to a solution in another part of the puzzle (i.e. the other solution must be substituted for the numeral). Some numerals are not references, but distinguishing them programatically is not straightforward, so we omit them. See for example, the substitution required above of "Scottish" for "20".

4. 90 clues that do not match our regular expression or with an empty clue after regular expression extraction.

5. 56 where the answer does not match the length enumeration.

6. 85 where there are unrecognized characters in the clue (e.g., unparsed HTML).

We further remove 1,611 exact duplicates. These are clues with the same target answer and clue strings that match after lower-casing, normalizing whitespace, normalizing articles ("a", "an", "the"), and stripping off punctuation.

In addition to releasing the full dataset, code to fully replicate our data download and pre-processing pipeline is also available in the GitHub repository. The code we provide reproduces this detailed information, including removal counts broken down by reason, whenever it is run to generate the data splits; comments in the code provide additional details.

## C   Baseline experiment details

We provide details of model and task set-up, hyperparameter choice, machines and compute used, and evaluation methods.

Evaluation is the same for all models. When evaluating the correctness of outputs, we lowercase all letters and ignore whitespace. Generated whitespace (i.e., spaces between generated multiword answers) is considered only for evaluating the meta-properties (e.g., number of words) for model outputs. The GitHub repository includes code to exactly replicate all evaluations.

### C.1   WordNet

The WordNet heuristic approach produces candidate outputs as follows: the first and last words of a clue are extracted from the clue and lowercased. For each of these two words, we do a reverse dictionary lookup using WordNet. We try building the reverse lookup with synonyms, hyponyms, and hypernyms, where the last two have controllable lookup depth (e.g., hypernyms of the first set of hypernyms, etc). Any underscores or hyphens in WordNet lookup results are replaced with spaces. We test with and without inflection of lookup results by using [19] to produce all possible inflections. We filter to outputs of the correct length, excluding whitespace. We try ranking outputs (1) by by their multiset character-level overlap with the rest of the clue (i.e. not the word used for the reverse lookup), (2) by bigram overlap with the rest of the clue using a modified Levenshtein distance, and (3) by the order in which they are added to the output set (i.e., without further ranking). For this model, the number of generated outputs is determined by changing which parts of the WordNet graph (synonyms, hyponyms, hypernyms, and depth) we use to generate candidates.

This model does not involve any training, so the train set is not used. We take the configuration that produces the best performance on the dev set: we use reverse lookup with synonyms and hyponyms to depth 1, omit inflected forms, and rank using multiset character-level overlap.

We can upper-bound this method by observing that, when including synonyms and hyponyms/hypernyms up to depth three, and inflecting all outputs using LemmInflect [19] (i.e., producing the maximum number of candidates for each clue), our definition sets contains the correct answer 22% of the time. This performance could be achieved if we had a perfect ranking function. However, since our ranking mechanism is poor, we do not achieve this level of performance and find that the best outcome is achieved by reducing the size of our reverse dictionary space to include only synonyms and hyponyms to depth 1.

The slowest of these models is the one with full hyponym/hypernym lookup to depth 3 and was run on a 2013 Macbook Air in two minutes.

## C.2 KNN BoW

The KNN model is implemented with scikit-learn's [28] CountVectorizer and KNeighborsClassifier. The CountVectorizer lowercases all characters and considers only alphabetic characters, numbers, parentheses and the | character. All other characters function as split locations and are themselves omitted. When including the length enumeration we append length as, e.g., *(4)* or *(4|6)*, in the case of multiword solutions. We use '|' so that the length enumeration is treated as a single token. As for all other traininable models, targets for the train set are lowercase solutions with spaces separating multiword answers. We select the 3000 nearest neighbors for each test clue so that we always produce at least ten outputs of the correct length for each clue.

We train by fitting the train set and take the set of hyperparameters that produces the best performance on the dev set: in particular, we use 1-grams, since performance degrades with longer n-grams.

This model was run on a 2013 Macbook Air in roughly ten minutes.

## C.3 Rule-based

We run the Deits [5] solver on our clue sets. The model is not trainable, so we directly evaluate it on our dev and test sets. We follow Deits' guidance to set up our clue file, providing a text file where each line is of the form, *clue | answer* – for example,

*But everything's really trivial initially for a transformer model (4) | bert*

We do not restrict the number of outputs generated by this model.

The rule-based solver uses a context free grammar (CFG) that specifies possible clue forms. For example, a grammar for an anagram clue type could be "$Anagram $AnagramIndicator $Definition". Terminals for $AnagramIndicators (and other types of indicators in the full grammar) come from custom lists of indicators. One of the components of the CFG is a definition: the definition terminal is matched to a word or set of words. The non-definition part of the grammar ("$Anagram $Anagra-mIndicator" in the above example) is evaluated to produce possible wordplay outcomes (in this case, computing valid anagrams of the tokens matched to the $Anagram terminal). Finally, the possible wordplay outputs are compared to the definitional tokens using WordNet's word similarity function. Parses with higher similarity are ranked higher.

As mentioned in the footnote in Section 4.3, Deits [5] has a more recently implemented solver that is reportedly faster. Because the Python solver is slow, we set a timeout of 120 seconds (timed-out runs usually still produce some candidate answers) and report an average time to solve a clue of roughly 40 seconds. This results in a total time to evaluate each of the dev and test sets of approximately 300 CPU hours. We evaluate this model using multiple internal cluster CPUs run in parallel.

## C.4 T5: vanilla seq2seq

Starting from HuggingFace's [39] pretrained model parameters, we fine-tune T5-base to produce the target answer for each clue input. As described in Section 3.1, inputs are of the form, e.g., *But everything's really trivial, initially, for a transformer model (4)*, with output *bert*.

We optimize with Adafactor [33] using the relative step and warmup initialization options, as implemented in the HuggingFace library (all other parameters are left unchanged from the HuggingFace defaults). We use a batch size of 256 input–output (clue–answer) pairs with per-batch truncation-to-longest, which is implemented by HuggingFace's T5FastTokenizer. We train with a patience of 15 epochs and select the best model according to dev set performance, based on whether the top answer (over 5 beam search generations) is correct. During evaluation, we generate 100 outputs for each input (100 beams with 100 output sequences) in order to evaluate sample metrics. Hyperparameters, including those for generation (max-length=10 tokens, length-penalty=0.05), were selected based on dev set performance. This setup, including all hyperparameters, is implemented in the code that we release on GitHub.

We use an internal cluster. Training takes approximately 100 minutes on a single GeForce RTX 3090 GPU. Evaluation takes roughly 120 minutes.

# D Curriculum learning

## D.1 Datasets

### D.1.1 ACW-data

ACW-data is the unprocessed version of the American crossword clue dataset [29]. To preprocess it, we

1. Remove clues that do not match our reverse-dictionary goal: We remove clues that contain underscores or multiple hyphens, since these are generally fill-in type clues, rather than phrasal synonyms. We remove clues that reference other clues, i.e., those containing "Across" or "Down" in the clue text. We remove clues likely to be abbreviations, i.e., those with a clue ending in a period with an answer fewer than four letters, since cryptics rarely include abbreviations. We remove clues where the clue text ends in a question mark.

2. We attempt to make the clues resemble our dataset by removing any periods that occur at the end of clues, since cryptic clues do not generally have periods at the end of normal clues (though they do admit other types of punctuation).

3. We filter normalized duplicates using the same approach as for cryptic clues (i.e. clues with the same clue and answer strings after normalizing case, whitespace, and articles and stripping punctuation.

This produces a cleaned dataset of 2,464,951 clue-answer pairs from which we produce the three ACW-data-derived datasets used in curricular training. It is worth noting that some of the answers in this dataset are multiword answers that are unsplit. Optimally we would find a way to split these answers to increase similarity to our primary dataset, which does split multiple word targets.

The code to reproduce this preprocessing and to produce the following datasets is included in the GitHub repository. Details of the three datasets (ACW, ACW-descramble, and ACW-descramble-word were given in the main paper (Section 5.1).

### D.1.2 ACW training datasets

The actual input-output pairs for ACW, ACW-descramble, and ACW-descramble-word are produced from the processed version of ACW-data at train time. At train time, we prepend a task label as described in the main text. The ACW curricular dataset has no further modification. For ACW-descramble and ACW-descramble-word, we produce a scrambled version of the letters during dataset collation and modify the input as specified in the main text. The provided code includes the collation functions that produce the final input–output pairs for these three datasets.

### D.1.3 Anagrams dataset

First, we produce a list of valid English words to be considered for anagramming from a publicly available dictionary of English words. Using this list of words, we group all words into whether they are anagrams of each other (i.e. grouping them by their sorted letters). For anagram indicators, we use Deits [5] list of anagram indicators.

This produces 13,535 anagram groups (i.e., 13,535 unique sets of letters from which can be realized at least two valid English words). These groups contain a total of 32,305 total words. The anagram indicator list has 1,160 anagram indicators. At train time, a curricular epoch consists of showing each anagram group to the model once. To do this, during collation at train time, we randomly sample two of the anagrams from each set, randomly sample an anagram indicator, and randomly sample a position (prepend or append).

## D.2 Training

As described in Section 5.1, each supplementary dataset has its own task label [31], which is passed to the model as part of the input string, and all inputs include length enumeration as in the vanilla T5 case. We fine-tune T5-base in the same way as described in Appendix C.4, but with the following modifications.

For curricular training, we first fine-tune on one or more supplementary tasks according to a training schedule, for which we tune the following hyperparameters: the number of curricular epochs, the frequencies with which each task is shown, whether the Adafactor optimizer is reset before primary training (only affects non-constant LR, i.e., when we are training T5-base but not when we are training T5-large), and the frequency of curricular subtask review during primary training. We hand-tune these hyperparameters, generally finding that training nearly to convergence on a held-out dev set for the primary curricular task is optimal. We also find that, for T5-base, resetting the optimizer between curricular and main training slightly improves performance. The specific configurations to replicate curricular training are included in the GitHub repository.

In order to directly compare the different curricula, we set up the curricula so that the number of training examples shown to the model in each epoch as well as the mix between curricular and primary task are the same. For example, for our single-dataset curricula (ACW and ACW-descramble), we run experiments with 4 curricular epochs and relative batch frequences (primary dataset: curricular dataset) during main training of 20:6. When training on curricula that include two curricular datasets, we do only 2 curricular epochs and use relative batch frequencies of 20:3:3 (primary: curricular 1: curricular 2).

To produce Table 3a, we evaluate only on the dev set over five generations to enable faster iteration. To produce the second column of the table, we algorithmically identify anagram clues. Code to replicate the anagram labeling and evaluate on this subset is available in the GitHub repository.

To produce our top result in Table 2, we double the total number of curricular epochs (from 2 to 4), select the best model checkpoint via dev set performance, and perform final evaluation on the test set taking 100 generations.

For all curricular training we use an internal cluster. Each curricular epoch takes roughly 150 minutes, giving a total curricular training time of roughly ten hours. Primary training afterward takes roughly 130 minutes since we continue to review the curricular datasets. This gives a total train time of roughly 12 hours on a single GeForce RTX 3090 GPU.

# E  Model analysis details

## E.1  Descrambling task

We start with the preprocessed version of ACW-data from Appendix D.1.1 and further remove any clue–answer pair with an answer that is not in an English dictionary (e.g., multiword answers would be removed). This guarantees that all descrambling targets are valid English words.

After removing multiword answers, we have a dataset of 1,796,078 clues. We keep only words that have between 4 and 14 characters and downsample to 10% (roughly 180k clue-answer pairs).

We train T5-base to complete the descrambling tasks using the same approach as in Appendix C.4. Code to replicate dataset creation, training, and evaluation are available in the GitHub repository.

## E.2  Wordplay systematic learning

Detailed code that identifies first name anagram substrates and generates substitutions is included in the GitHub repository. For name identification, we use names lists from the US Naval Academy and the U.K. Office of National Statistics (both lists, including with download URLs are provided in the GitHub repository). We identify 27 clues in the dev set and 69 clues in the train set that require anagramming a single word that is also a first name, and for each we perform 10 scramble and 10 name substitutions.

## E.3  Efrat et al training

We use the same training setup as in Appendix C.4, but with the following changes: we train T5-large with a constant learning rate of 3e-5 and an effective batch size of 768. For evaluation we use the same metric (top output with b=5 beams, no filter) as used by Efrat et al. [7].

We again train on an internal cluster using a single GeForce RTX 3090 GPU. Training to replicate Efrat et al. [7] results (i.e. non-curricular) takes roughly ten hours. Curricular pretraining is done for

3 epochs and takes roughly 4 hours per curricular epoch, giving a total time for curricular pretraining of roughly 12 hours.

Code to replicate this approach is included in the GitHub repository.