# OpenReview forum: "Decrypting Cryptic Crosswords: Semantically Complex Wordplay Puzzles as a Target for NLP"
_NeurIPS.cc/2021/Conference — NeurIPS 2021 Poster_

### Official Review · Reviewer_oTbN · 2021-06-27

**Rating:** 7
**Confidence:** 4

**Summary:**

Authors 1) provide a dataset of cryptic crosswords; 2) explore baselines performances on this dataset; 3) provide interesting insights on those performances and come up with a pretraining strategy that increases performances.

Briefly, cryptic crosswords are puzzles that require both mastering definitions and wordplay (i.e. reordering of letters, extracting letters, composing parts of several words to form a novel word, etc.) to be solved. Authors provide a large dataset of such puzzles (in english), that they introduce as a stepping stone to solve more complex real-life settings.

They show that carefully crafted non-neural baselines are not very good (either rule-based following previous work on this, or KNN) and that even T5 (state of the art text-to-text transformer pretrained on many tasks, such as translation, summarization, etc.) cannot be finetuned for optimal performances on this dataset. Worst, when considering adversarial splits, such that for instance all puzzles with answers starting with the same subword are found in the same split, T5 performances drop dramatically.

They hint at a solution for this problem via curriculum learning, and devise the following strategy: first pretrain T5 on a series of tasks resembling the final wordplays (e.g. unscrambling a word using its definition: "etalp + flower part" --> "petal") and only then finetune T5 on the actual cryptic crosswords dataset. T5 performances indeed increase, but authors note that they are far from human performances, or even acceptable performances for a machine.


**Limitations And Societal Impact:**

Authors have addressed most limitations of their work. It does not seem to have negative societal impact.

**Main Review:**

Globally, this paper is easy to follow, and explains each step with enough details so that it may be understood without problem. It should also be noted that authors have made strong efforts to be honest about their work: they explicitly state that they do not claim to solve the difficult task they introduce, rather provide insights into promising future work and, very importantly, compare themselves candidly to the contemporary and concurrent work of Efrat et al. 2021 -- which also introduces a cryptic crosswords dataset.

First of all, the strengths of the paper:
- As I said, paper is well crafted and is easy to read and understand.
- The dataset introduced is relevant to the NLP community and seems like a good diagnostic dataset for methods that should be able to deal with subtle variations in language (simulated by wordplays).
- Contrary to the work of Efrat et al. 2021, which introduces a similar dataset andalso uses a finetuned T5 baseline, authors of this paper also propose credible strategies to improve performances, via curriculum learning.
- Experiments done are sound, and convincing. In particular, the new split introduced is clever, and show efforts in understanding where T5 might shine by chance, rather than by skill.

Despite those strengths, the paper suffers from several flaws, which are difficult to overlook:
- The related work section is somewhat light: 10 lines in total, and only refers to cryptic crosswords. I understand that space is tight in conference papers, but authors should also refer to work related to other areas of this work, for instance curriculum learning using similar, simpler tasks. For instance, Wu et al. 2021 [1] and Geva et al. 2020 [2] are two different works that explore the same strategy but for numerical reasoning.
- Similar to the related work focusing only on crosswords, the abstract and introduction seldom mention the usefulness of solving this specific task, for real world applications. Despite authors claiming to "characterize the problem and motivate its importance", motivation is rather lacking except for a few lines (e.g. in the abstract "NLP systems that seek to process compositional language in more creative, human-like ways") which need argumentation.
- While the rule-based and T5 baseline are relevant, there still seem to be lacking a crucial baseline: any character based model, which we would intuitively think has a better chance at rearranging letters than subword-based models such as T5.
- There are no experiments on other data (such as maybe tweet sentiment analysis?), that would showcase the presented improvements as useful in real-life settings.

EDIT: after a response from the authors, I'm updating my score from 6 to 7: Good paper, accept.

All in all, I think this paper is a marginally above acceptance threshold, because many of its flaws can be seen as future work and do not cripple the work as a whole. The dataset is great, and in my opinion relevant to NLP community because it introduces complex language manipulation. The experiments and claims are sound, well-designed and well explained. However, I feel that it could be argued that the methods presented here would be better suited for Empirical Methods in NLP rather NeurIPS. Furthermore, crucial experiments, either by using a character-based baseline, or a real-life dataset, are missing. This problem is attenuated because introducing a dataset, its scope and analyzing some baselines is in itself a great piece of work. (I also don't count as a negative the concurrent work of Efrat et al. 2021, which is too recent). I strongly feel that at least the related work should be reworked to include more than cryptic puzzles.

If this paper were to be rejected, I would advise authors to carefully reframe the paper, so that the importance of the presented tasks is better highlighted. For instance, the abstract should start by introducing complex language manipulation tasks as a real world challenge, and then present cryptic crosswords as a relevant diagnostic dataset, rather than focus directly on crosswords. Similarly, introduction and related work should also focus on related areas (e.g. curriculum learning, tasks with fuzzy nlp, etc.), which are arguably more important than simply solving crossword puzzles. I would also include stronger baselines (character based models) and real-life experiments.


[1]: LIME: Learning Inductive Bias for Primitives of Mathematical Reasoning -- Wu et al. 2021

[2]: Injecting Numerical Reasoning Skills into Language Models -- Geva et al. 2020

**Time Spent Reviewing:**

6

---

> ### Author Response · Authors · 2021-08-09
> **Response to suggestions from Reviewer oTbN**
>
> We appreciate this thoughtful review and summary, which we respond to below.
>
> # Related work
> We appreciate the suggestion to reference related work outside of the cryptic crossword domain and will substantially beef up the related work for the CR using the extra page. In particular, we will  discuss how our work relates to curriculum learning and numerical reasoning, as well as to common NLP tasks that might benefit from our approaches like question-answering.
>
> # Related tasks
> We agree with the reviewer that we could better motivate the relevance of this task to other common NLP tasks. We appreciate the suggestion to reframe the introduction to focus on the domain’s relevant to other common NLP and ML tasks (as well as more broadly to cognitive study)! We cut some of this related work discussion  due to space constraints: we thought that most readers would be unfamiliar with cryptics and that clearly framing the domain needed to take precedence over relating it to other areas.
>
> We think that, insofar as the task requires combining a flexible understanding of linguistics and metalinguistics with structured semantic parses, it could be widely relevant for any task that requires similarly rich semantic parsing in structured domains (e.g., solving logic puzzles, chatbots that navigate structured spaces, natural language knowledge/database queries, natural language specified computer programs, etc.). Hopefully inclusion of this discussion in the CR can to some extent address the reviewer’s concern that we need to introduce complex language manipulation as a relevant, real world challenge!
>
> # Character-based models (and wordplay)
> We agree with the reviewer’s intuition but want to point out that, to some extent, T5 can do character manipulation and does exhibit length understanding! Given that we shared the reviewer’s concerns, we studied the limitations of SentencePiece tokenization in Sections 6.1 (meta-linguistic properties like length) and 6.3 (character level manipulations like descrambling) and uncovered interesting evidence that T5 *can* actually handle certain kinds of character-level manipulation. This suggests that T5 may actually be better suited for the task than it first appears. Given T5’s relative success on the experiments in T5 as well as an additional unreported experiment that we discuss below, a purely character-level model might not actually be the right choice here.
>
> ## T5 can descramble letters
> First, Section 6.3 shows that T5 can do word descrambling under a random split (64/92% top1/top10 performance). If we take these as rough performance ceilings, our best curricular approach (20/42% respectively) still has substantial room to improve. A possible counterargument is that there is less headroom on the disjoint split: (12/24% performance on descramble vs 6/20% with our best curricular model).
>
> ## T5 can interpret length constraints
> Second, in Section 6.1, we show that T5 does a reasonably good job of learning the correspondence between length enumeration and actual word length -- i.e. the sum of lengths of constituent SentencePiece tokens -- (48% of top 10 outputs have the correct length). This suggests that T5 both can “understand” the length-enumeration part of the input and also that it learns the interaction between lengths of different tokens in a word with the length enumeration.
>
> ## Character-based models
> Lastly, we ran some unreported experiments in which we attempted to provide character-level information to T5, for example, by putting a space between each character or providing each word with and without spaces. We saw that the model achieved similar performance to the baseline with these modifications. So, given that the results mentioned above do provide evidence that T5 can understand lengths and constituent letters, we did not pursue these methods further.
>
> Still, we agree that a character-based approach could further improve performance on our new benchmark, and it is a further research avenue that we are interested in. The success of subword tokenization (bytepair, SentencePiece, etc) schemes have generally been found superior to character-based representations. This suggests that a hybrid approach might actually be appropriate: a large pretrained model with subword tokenization can discover overall semantic structure and then access or manipulate character-based representations. Alternatively the idea could be tweaked to favor span-based identifications (which parts of the clue are the letters that compose the answer) combined with a decoding system that turns a set of letters into an output that matches the definitional embedding. All of these are exciting avenues for future work. Since it wasn’t immediately obvious to the authors whether such character-level study would have broader application, we briefly frame our justification for its relevance: first, breakdowns of “objects” to their constituents can be argued to be necessary for arithmetic/mathematical work (“character” level). Second, hybrid approaches in which a language model interacts with another system (database, symbolic reasoning) also involve exposing a secondary representation system and designing an interface between them.
>
> # Relevance to other “real-life” settings
>
> In addition to the connections to real-life settings mentioned above, we are confident that applicability to other NLP and ML settings will come out of future work as we and others make progress on solving this new benchmark task. Moreover, given space and time constraints, we thought it made sense to focus on the task we are introducing and our attempts to solve it, rather than applications to other domains. [We also note that, for some of us, crossword puzzles are part of real life! :-)]
>
> Many advances in ML have come from studying the games and puzzles that humans play: Consider the various videogames of early DeepMind work, and work on chess and Go. Friedman and Fine point us to the interesting cognitive nature of the “aha moment” in linguistic puzzles, and we hope that future research on this benchmark, including further human evaluation, may find useful links between cognitive study and the ML approaches to this task. In this way, this task can be argued both to be a useful diagnostic for related tasks, as well as a benchmark valuable in its own right.
>
> # Additional comments
> Regarding EMNLP vs NeurIPS: We considered submitting to an NLP conference, but our hope is that the general problem-solving nature of the task may attract interest from non-language ML researchers at NeurIPS, and we are eager to have that perspective.

---

> > ### Comment · Reviewer_oTbN · 2021-08-24
> > **Response**
> >
> > Thanks for responding in such details.
> >
> > I will revise my score up to 7, given that a lot of this discussion and discussions with other reviewers can be used to improve the paper for the final version, especially its positioning and the discussion around character based models.

---

### Official Review · Reviewer_T9VZ · 2021-07-15

**Rating:** 5
**Confidence:** 5

**Summary:**

- introduces a dataset comprised of cryptic crossword clues, and shows performance of a few baselines including a T5 prefix-LM are not great, and suggests the problem is hard for current models
- discusses use of related datasets/tasks and data augmentation to improve T5 baseline


**Limitations And Societal Impact:**

Limitations of T5 sentencepiece model are not sufficiently discussed for this character-based task.

I don't see any potential negative societal impacts.

**Main Review:**

Language needs harder tasks/benchmarks and this paper attempts to introduce one. The most important question to answer in the paper is is the problem difficult enough and unsolved to be a new challenge benchmark. I do not think this is answered by the experiments. The fact that a T5 base model doesn't do well is a good sign, and intuitively seems like a very hard problem. However, this particular baseline may not be well-suited to the problem since it tokenizes text into SentencePiece tokens, which makes it difficult to interpret character-length constraints given as input. To do this task properly, the model must learn the number of characters in each token, which presumably requires a lot of data, and may partially explain the poor performance. This limitation is not discussed in the paper. I'd like to see a baseline (preferably a pre-trained LM similar to T5) that takes in characters as input. It'd also be useful to see whether larger T5 models do better.

The "Curricular" model is really just supervised pre-training on related datasets (american crosswords) with some data-augmentation. It is not curricular learning, nor is it particularly novel.

In summary, there is not much in terms of new methodology, but the dataset is potentially interesting as new language benchmark. However, the experiments do not convince me that it is indeed hard enough to be one.


Questions/clarifications:

T5 prefix-lm is not the standard T5 model, which is encoder-decoder.
Why did you use the T5 prefix-lm model instead of encoder-decoder?


# Update after author response
Again I think this dataset is potentially interesting, but I'd like to see character-based model baselines as this task involves counting characters in the output which SentencePiece tokenization severely obstructs. It'd be great to see results in a future paper using something like ByT5. If a good character-based model cannot solve this task, I would be convinced that this is a great new benchmark for NLP; in the absence of that it is hard to tell. At the very least, some of the points in the author response should be added to the discussion in the paper. I've updated my score assuming this is added.

"This suggests that T5 both can “understand” the length-enumeration part of the input and also that it learns the interaction between lengths of different tokens in a word with the length enumeration."
- I'm sure it can but it must learn the length of tokens to properly do the task, which a model that decodes characters would have a much easier job doing.

Thanks for clarifying that the model is in fact an encoder-decoder.


**Time Spent Reviewing:**

1

---

> ### Author Response · Authors · 2021-08-09
> **Response to suggestions from Reviewer T9VZ**
>
> We thank the reviewer for these comments and suggestions. We address the reviewer’s primary concerns below:
> - Appropriateness of T5 and SentencePiece tokenization
> - Curricular approach and novelty
> - The difficulty of the task and whether it is “challenging”
>
> # SentencePiece tokenization and T5
> Regarding SentencePiece tokenization, we agree with the reviewer’s intuition that this is a possible limitation of the T5 model and acknowledge it in our submission checklist (1b). However, an important issue is that there are not good character-level models with the breadth and complexity of T5 to handle the semantic complexity of the cryptic domain. Moreover, given that we shared the reviewer’s concerns, we studied the limitations of SentencePiece tokenization in Sections 6.1 (meta-linguistic properties like length) and 6.3 (character level manipulations like descrambling) and uncovered interesting evidence that T5 *can* actually handle certain kinds of character-level manipulation. This suggests that T5 may actually be better suited for the task than it first appears. Given T5’s relative success on the experiments in T5 as well as an additional unreported experiment that we discuss below, a purely character-level model might not actually be the right choice here.
>
> ## T5 can descramble letters
> First, Section 6.3 shows that T5 can do word descrambling under a random split (64/92% top1/top10 performance). If we take these as rough performance ceilings, our best curricular approach (20/42% respectively) still has substantial room to improve. A possible counterargument is that there is less headroom on the disjoint split: (12/24% performance on descramble vs 6/20% with our best curricular model).
>
> ## T5 can interpret length constraints
> Second, in Section 6.1, we show that T5 does a reasonably good job of learning the correspondence between length enumeration and actual word length -- i.e. the sum of lengths of constituent SentencePiece tokens -- (48% of top 10 outputs have the correct length). This suggests that T5 both can “understand” the length-enumeration part of the input and also that it learns the interaction between lengths of different tokens in a word with the length enumeration.
>
> ## Character-based models
> Lastly, we ran some unreported experiments in which we attempted to provide character-level information to T5, for example, by putting a space between each character or providing each word with and without spaces. We saw that the model achieved similar performance to the baseline with these modifications. So, given that the results mentioned above do provide evidence that T5 can understand lengths and constituent letters, we did not pursue these methods further.
>
> ## Additional remarks
> We recognize that our mention of SentencePiece comes quite late in the text (Section 6.3) and for CR will mention in the Introduction this possible limitation and also that we study this potential limitation in our experiments in Section 6.1 and 6.3 (which *do* demonstrate that T5 is in fact quite robust to character-level challenges!).
>
>
>
>
> # Curricular approach and novelty
>
> ## Curricular learning
> We use “curricular” here in the sense originally used by Elman’s “starting small” work, cited as foundational by Bengio et al. in their seminal paper on the topic (Bengio, Y., Louradour, J., Collobert, R., & Weston, J. (2009, June). Curriculum learning. In Proceedings of the 26th annual international conference on machine learning (pp. 41-48)). We acknowledge that this differs in some respects from “curricular learning” as used today (where the curriculum itself is often learned as opposed to feeding a fixed curriculum)  and will make this distinction in the CR.
>
> ## Novelty
> We will also clarify what we see as the novelty of the paper. We do not claim that our approach (which involves pre-training on related data sets with augmentation) is unprecedented. The novelty is in the dataset (characterizing, curating, and  releasing it) and in developing a novel approach for the problem that demonstrates considerable improvement over the baseline. As discussed in the paper, the results are interesting and unexpected in various ways and so we hope this work will be of interest to the community, spurring future work on this benchmark, with potentially broad impact.
>
> With respect to curricular learning or “pre-finetuning” contributions, we show what kinds of training schemes work best, how they interact with disjoint data sets, etc. In particular, we demonstrate the potential effectiveness of curricular learning approaches for the cryptic domain with a very simple “curriculum”. Given the high degree of variability in cryptics (i.e differing levels of compositional complexity), we believe that further work on this benchmark may benefit, for example, from automatic tuning policies. Moreover, the high degree of compositionality in this benchmark task and the usefulness of the synthetic datasets we produce may also provide an opportunity to advance existing curricular approaches. Regarding new methodology: we hope the reviewer will consider that, beyond demonstrating a new method for improving performance on this dataset, that Section 6 presents numerous useful contributions, including a word-initial disjoint split, a study of a baseline wordplay task, and a study of the extent to which the model learns the systematic nature of the clues. These approaches can be used to study the extent to which other architectures generalize on this task.
>
> # Task difficulty
> To be honest, our original worry was that the task would prove too hard, not that it is not hard enough to be a language benchmark! Our results reassure us that the problem is tractable, but the current numbers also suggest that some serious innovations might be required to truly approach human performance. We hope that the divergence in opinion as to the difficulty and scope of the task is some evidence that this is a fascinating area worth studying!
>
> # Additional comments
> In terms of whether larger T5 models do better, our paper does show the difference between T5-base and T5-large. Since T5-large achieves good performance on a large set of NLP tasks, and since we sometimes characterize this as “language understanding,” we might expect similarly good performance on this benchmark if the model does truly “understand” language.
>
> Finally, we thank the reviewer for noting that the prefix-LM version of T5 is non-standard. This was simply a mistake in writing and we did in fact use the normal encoder-decoder architecture. (We have verified this in our code and will correct the error in the text.)

---

### Official Review · Reviewer_zZBA · 2021-07-18

**Rating:** 8
**Confidence:** 3

**Summary:**

**What is the task?**

Cryptic crossword puzzles task

**What are the contributions of the paper?**

* Presented a dataset of cryptic crossword clues as a challenging new benchmark for NLP systems that seek to process compositional language in more creative, human-like ways.

* A novel curriculum approach, in which the model is first fine-tuned with related, synthetic tasks (e.g., an augmented word descrambling task) before tackling actual cryptic clues.

* Introduced a challenging word-initial disjoint data split to test composition and generalization and investigated model behavior by systematically perturbing the wordplay part of clues, showing that the model exhibits behavior partially consistent with human solving strategies.


**Ethical Concerns:**

No ethical concerns

**Limitations And Societal Impact:**

Author(s)  discuss  opportunities to improve our approach and the limitations of using subword-level tokenization on a wordplay task.

**Main Review:**

**What has been done before?**

Efrat et al. —concurrent work that presents a similar dataset and similar neural baseline using T5.
While Efrat et al. [6] conclude that train/test disjointness is important, they do not fully consider T5’s robustness to plural and other inflections. The word-initial disjoint split that presented in the paper addresses this. The proposed curricular approach outperforms Efrat et al. work on different kinds of splits, especially word-initial disjoint split.

**What are the main results?**

* Results show that a curriculum of synthetic datasets can address this, substantially improving performance on the primary cryptics task.

* The proposed curricular approach outperforms previous best work on different kinds of splits, especially word-initial disjoint split.


** Are others (researchers or practitioners) likely to use the ideas or build on them? **

This curricular approach may be useful in other settings where focused linguistic capabilities are required. Cryptic crossword task can serve as a good benchmark for those interested in building NLP systems that can apply linguistic and meta-linguistic knowledge in more creative, flexible, and human-like ways.



**Strengths:**

* In this paper, author(s)  aim not only to present a data set and propose a novel solution but also to characterize the problem and motivate its importance.

* The claims are well supported by experimental results and look reproducible.

* The proposed curricular approach outperforms previous best work on different kinds of splits, especially word-initial disjoint split.

* Other researchers or practitioners likely to use the ideas or build on them.

Weakness/ concern:

* No formal comparison with human performance on the task


**Time Spent Reviewing:**

1.5

---

> ### Author Response · Authors · 2021-08-09
> **Response to suggestions from Reviewer zZBA**
>
> We thank the reviewer for these comments. We address the reviewer’s primary concern about comparison to human performance:
>
> # Human comparison
> Regarding formally comparing with human performance: We agree that human evaluation would be a great way to augment this research. (In fact, we are planning data collection for such an experiment!)
>
> Friedlander & Fine (psych researchers, citations [9] and [10] in our manuscript) have some interesting work exploring cryptic solving for both experts and novices and elucidate a number of interesting ways in which cryptics draw on interesting problem-solving and linguistic ability.

---

> > ### Comment · Reviewer_zZBA · 2021-08-25
> > **Response**
> >
> > Thanks for your response.

---

### Official Review · Reviewer_toeW · 2021-07-20

**Rating:** 7
**Confidence:** 4

**Summary:**

This paper presents a new dataset of cryptic crossword clues. The work presents an extensive evaluation study on this dataset with 3 non-neural baselines and one neural baseline, namely the T5 transformer-based model. The authors investigate as well a kind of pre-finetuning stage (a phase of finetuning before the final finetuning stage on the main dataset) as a way to improve the performance of the model on this task.

Contributions:
New dataset of cryptic crossword clues that has the potential to foster more research on complex reasoning tasks

Strengths:
- The writing is compelling and the paper is a very enjoyable read
- The author do a detailed analysis to understand better the limitations and abilities of transformer-based models on the task, including interesting ways of looking at data splits
- The authors investigate the use of a pre-finetuning stage as a way to improve the performance of the T5 model on the task
- The evaluation is convincing with multiple baselines, non-neural and neural.

Weaknesses:
Sometimes, the paper seemed to be a bit condensed

**Limitations And Societal Impact:**

The authors do not address the limitations and potential negative societal impact of their work.

**Main Review:**

Intro:
Lines 26-27: What does flexible mean in flexible AI or flexible NLP models?
Line 82: in what sense is it competitive? (Since you don’t have access to it to test it)

Dataset:
Is there a reason why the authors did not collect a “clue type” for each entry so as to potentially train models on that additional dimension vs without and see how that affects the learning?

Section 3.2:
Thoughts on formulating the problem similar to a MCQ problem where the answer is one of several choices?

Section 3.3:
- When determining if the answer is correct, how are the generated and golden answers compared? For example, is the generated answer lemmatized etc?
- Clarify more the following: “For instance, the best-performing crossword solver for American-style (non-cryptic) crosswords relies heavily on satisfying grid constraints.”

Section 5.1:
- The 7 in (7) for ACW, ACW-descramble, ACW-descramble-word is supposed to be the number of letters in the right answer, right? So I’m supposing this is a typo?
- It would be good to explain the other anagram indicators in the section so that the reader has a better appreciation of the dataset at hand. At the very least this could be done in the appendix which is not the case.

Section 5.3:
Some aspects of your work remind me of the Muppet paper from FAIR (Aghajanyan et al 2021). Your curriculum learning phase could be seen as analogous to their pre-finetuning phase and the overall theme is not far (e.g. some of the downstream tasks there could be commonsense reasoning or NLI). What’s interesting is that they found in almost all cases (datasets/tasks) the performance was hurt when only a few tasks were used for pre-finetuning and typically it’s when they pre-finetuned on more than 15 tasks that they began to see a consistent improvement on the final task. Your findings seem to be in contrast to that (i.e. with only a couple tasks for pre-finetuning you got an improvement), so I was wondering if you had any thoughts on that.

Section 6.2:
It would be interesting to assess quantitatively the similarity between different clues linked to the same word so that we can actually see whether the results are actually due to said similarity and not for some other factor. That is, it could be that different clues could have the same answers but not necessarily a high lexical overlap.

Typos/Minor comments:
Line 62: missing [ ]
Line 63: data set -> dataset
Line 65: finetune -> finetuned
Line 66: It would be good to add a sentence or 2 in the intro to clarify more (at this point) what you mean by disjoint data split so that the reader has a clearer idea about the different contributions of the paper from the intro
Line 329: we we
Lines 359-360: Incomplete bibliographic entry

Presentation comment:
It’s somewhat annoying that Tables 2 and 3 appear quite far from the actual relevant sections. You can consider referring to the relevant section in Section 3.3 and keeping the first instances of table citations to Section 5 so that they appear closer to the section.

Missing references:
- Line 155: Add the same reference [28] to the C4 Corpus (if some reader is not familiar with T5, they might not know directly that the corpus was introduced in the same paper)
- Generally, the Related Work section does not cover any work that involves this idea of (pre-)finetuning on another task before the actual finetuning phase. Some papers that do that: Tafjord et al. 2019, Shwartz et al. 2020, Aghajanyan et al. 2021

General feedback:
The main contribution of this paper seems to me to be the presentation of a new dataset which, I believe, is important. The main ML component of the paper is essentially the curriculum learning approach. As I mentioned, similar ideas of pre-finetuning have been introduced in at least 3 papers that I am aware of, and so, there's not much novelty here in my opinion (despite the specifics of the method adopted here). The question is then whether this paper is a better fit for an NLP conference. I am still of the opinion that the paper has a valuable contribution and so giving it an accept.


**Time Spent Reviewing:**

4.5

---

> ### Author Response · Authors · 2021-08-09
> **Response to suggestions from Reviewer toeW**
>
> We thank the reviewer for this thoughtful commentary and the great connection to recent work on pre-finetuning. We more or less follow the order of the reviewer’s “Main Review” section below.
>
> # Flexibility (lines 26-27)
> “Intro: Lines 26-27: What does flexible mean in flexible AI or flexible NLP models?” We mean “flexible” in the sense that the system can adapt to subtle variations and new inputs. That is, we don’t want to just write down all the possible cryptic crossword clue function possibilities (since doing so would be impossible) but want to ultimately build a system that can “flexibly” adapt. Such flexibility is often written about in the cognitive science literature as a hallmark of human intelligence.
>
> # Crossword Genius app
> “Line 82: in what sense is it competitive? (Since you don’t have access to it to test it)” The app’s creator was willing to run his system on our dataset, but he was not willing to share how the system was built and trained nor whether any of the clues had been seen before.
> He reported that the app achieved top-1 performance on par with T5-large under a naive split (49.1% for his system vs 56.2% reported by Efrat and 53.2% by us). However, our inability to characterize how the system is built means that we can’t present the app’s approach as a baseline: we can’t characterize the method scientifically, and we do not have a way to explore whether it generalizes. Most problematically, we suspect many of the clues in our data set have already been absorbed by the app but cannot confirm that since the method and data used by the app are not open. We hope that the app’s creator will, at some point, share his method with the academic community.
>
> # Dataset: clue types
> “Dataset: Is there a reason why the authors did not collect a “clue type” for each entry so as to potentially train models on that additional dimension vs without and see how that affects the learning?”
> This is an interesting avenue that we explored, but it is actually quite challenging to produce clue type labels for each entry (also remember that many clues can involve multiple manipulations). If we could do this in general, that would likely be of use to our solver. We were able to algorithmically label some clues with a bit of clever coding, but it was a relatively small subset. We used the anagram labels to assess how curricular training changed performance on this anagram subset. A possible further avenue for work on this benchmark task would be to provide additional supervision in the form of such clue-type labels or full parses.
>
> # MCQ formulation
> “Section 3.2: Thoughts on formulating the problem similar to a MCQ problem where the answer is one of several choices?”
> This is a good idea for further research. One challenge would be selecting appropriate distractors that enable us to study how the model succeeds, fails, and generalizes. Perhaps the way in which this would be most informative is to see if it addresses at all the underperformance of the model on the disjoint split, since train and test outputs would be the same (i.e. just numerical selectors, rather than actual words).
>
> # Model evaluation (answer comparison)
> “Section 3.3: When determining if the answer is correct, how are the generated and golden answers compared? For example, is the generated answer lemmatized etc?”
> The generated answers are lowercased and whitespace is removed (golden answers are preprocessed to have this form, too). We do not do any lemmatization here (though we do use lemmatization in the WordNet baseline).
> These details are also included in Appendix B, and the code supplement implements these behaviors. Also, as we note in Appendix B, generated whitespace is considered only for evaluating the number-of-words metaproperty of outputs but not for gauging correctness.
>
> # Grid constraints
>
> Section 3.3: Clarify more the following: “For instance, the best-performing crossword solver for American-style (non-cryptic) crosswords relies heavily on satisfying grid constraints.”
> By grid constraint, we mean that, since answers to different clues overlap, having a tentative answer to one clue informs the answer to an overlapping clue. For example, a clue for “flower (4)” with third letter “s” is likely to be “rose” and not “iris”. Since nearly every grid position in an American-style crossword is part of two clues, grid constraints are heavily relied upon to solve puzzles. Cryptics, on the other hand, generally have very little overlap betwe en clues (one answer might intersect only one or three other answers), so the wordplay, rather than the grid, provides most of the information to “decrypt” a clue.
>
> # Minor comments on Section 5.1
> - “The 7 in (7) for ACW, ACW-descramble, ACW-descramble-word is supposed to be the number of letters in the right answer, right? So I’m supposing this is a typo?“
>
>     Yes, good catch - thank you! We switched the example from a word of length seven in a late draft.
> - “It would be good to explain the other anagram indicators in the section so that the reader has a better appreciation of the dataset at hand. At the very least this could be done in the appendix which is not the case.”
>
>     Thanks for this suggestion; we will add to the appendix further examples and details of anagram indicators, and do the same for one other indicator type.
>
> # Prefinetuning (Muppet paper from FAIR) Re. Section 5.3
> Thank you for pointing us to this relevant research. This suggests that it is worth trying a larger collection of different word manipulations in the pre-finetuning / curricular phase in future work on this benchmark. One of our unreported experiments involved pre-finetuning on both an anagram and hidden dataset. Although the two datasets individually boosted performance on their respective clue subsets, when used together, we saw degraded performance. Because of this we instead focused on data augmentation with phrasal definitions from the ACW dataset.
>
> We think our research may suggest that, in cases where the curricular or pre-finetuning tasks are not linguistically similar (e.g. a wordplay without a phrasal definition), it is difficult to coax the model to incorporate the learned transformation. This may be an important distinction with the Muppet paper: for many of their tasks, the model had a separate classification head. Given that clues do not, in general, have type labels and also that many clues require the application of multiple, successive transforms, we cannot provide multiple heads without considerable adaptation. This difference may point future researchers in another direction that we considered: a recursive routing network or reinforcement-learning based approach might be able to address the need for successive, discrete character-level transforms.
>
> # Lexical similarity Re. Section 6.2
> We agree with the reviewer’s intuition. We conducted some qualitative study by manually reviewing clues with the same answer to see if they often include similar words. The answer was roughly that they do. Moreover, we think that the KNN BoW approach is a good quantitative test of this hypothesis: KNN can succeed only if clues with similar answers have higher lexical overlap with each other than with other clues. Given that KNN (lengths) can achieve 11.2% top-10 success, it does look like clue similarity could account for almost a third of T5 (lengths)’s success. Though we didn’t test it, we suspect that modifying the KNN method to handle lemmatization (e.g. so that the absence or presence of an ‘s’ is still considered correct) may further boost this performance, which would suggest that clues with similar but nonidentical answers are also similar.
>
> # Comments, presentation, references
> Thanks for catching our typos and the comment on table presentation. We’ll try reformatting and see if that layout works better. Thanks also for the helpful references related to pre-finetuning. We agree that we should we should situate our contribution in this set of prior work and will augment the introduction for the CR.
>
> # Conference type
> Regarding type of conference: we considered submitting to an NLP conference, but our hope is that the general problem-solving nature of the task will attract interest from non-language ML researchers at NeurIPS, and we are eager to have that perspective.
>
> # Regarding novelty of our curricular / pre-finetuning approach:
> We agree that some of the main contributions of this paper are the data set, framing of the problem, and general approach, and we are glad the reviewer appreciated those contributions and found them valuable.
> Regarding the novelty of the ML component: At least with respect to the Muppet paper, as we mentioned above, there are some interesting differences: for our task, we cannot use different task / classification heads as was done in Muppet. Moreover, our curricular tasks are actually something like disjoint sub-tasks of a larger compositional task, rather than related tasks that share problem-solving elements (i.e. the distinct tasks clearly draw on similar skills, whereas our sub-tasks are rather unique manipulations). Moreover, the interaction of the curriculum with the word-initial disjoint split is also new.
>
> # Regarding limitations and potential negative impacts
> We address limitations in the main text, regarding tokenization schemes and the nature of the task. We address negative societal impacts (which we believe are minor) in the checklist.

---

### Decision · Program_Chairs · 2021-09-28

**Decision:**

Accept (Poster)

**Comment:**

This paper presents a new task for evaluating language models, cryptic crossword puzzles, along with a new dataset to evaluate on this task. The authors also propose a curriculum learning approach that improves over a standard seq2seq baseline on the proposed task. Reviewers agree that this is a well written paper and the dataset is interesting to a lot of people. Reviewer T9VZ has a concern about adding a character-level baseline. I agree this would be something good to add, and I encourage the authors to explore this for the next version of the paper. I think the paper is above the bar for acceptance to NeurIPS, so I recommend acceptance.

**Consistency Experiment:**

NeurIPS has a long history of experimentation. In 2014, NeurIPS ran an experiment in which 10% of submissions were reviewed by two independent committees to quantify the randomness in the review process. This year, we repeated a variant of this experiment to see how the quality of the review process has changed over time.  This paper was part of the experiment and was therefore assigned to two committees (consisting of reviewers, an Area Chair, and a Senior Area Chair) that reached independent decisions.  If both committees made the same recommendation, this recommendation was followed. If a single committee recommended acceptance, the paper was accepted (with the exception of a few cases in which the other committee identified what we considered a fatal flaw, e.g., an error in a key result).

Both committees reached the same decision: **Accept (Poster)**

The other committee assigned to the paper recommended **Accept (Poster)**.  You can find the other set of reviews, along with any follow up discussion with the authors here:
https://openreview.net/forum?id=Ah5CMODl52